# Characterisation of the Surface Free Energy of the Recycled Cellulose Layer that Comprises the Middle Component of Corrugated Paperboards

János Keresztes [1] and Levente Csóka [2,*]

1 Faculty of Wood Engineering and Creative Industries, University of Sopron, 9400 Sopron, Hungary
2 Faculty of Informatics, Eotvos Lorand University, 1053 Budapest, Hungary
* Correspondence: csl@inf.elte.hu

**Abstract:** The objective of this study was to determine the polar and dispersive surface free-energy (SFE) components of the central layer of corrugated paperboards, which are made of recycled fibres. The polar and dispersive components of, and the total, SFE (also known as interfacial energy) were calculated from the contact angles of water and diiodomethane liquids on recycled cellulose sheets. The total SFE of the middle component layers of two different grammages which comprised recycled fibres ranged from 47.9 mN/m to 51.05 mN/m. The contribution of the polar component to the total surface free energy of the two types of sheets ranged from 8.6 mN/m to 12.6 mN/m. This polar contribution was significantly lower than that of water. The contact angle method proved to be a consistent way to estimate the surface properties of industrially made recycled paper products.

**Keywords:** surface free energy; polar component; recycled cellulose; contact angle

## 1. Introduction

The development of the use of recycled fibres in the global paper industry is a success story of the second half of the 20th century. Recycled fibres satisfy the need for fibres as a raw material in both industrialised and developing countries [1]. The fibres are produced through the collection and recycling of paper of different grades. Processing is a major challenge that is solved through the use of additives such as starch and clays [2] or surface treatment methods [3]. Several research groups have developed new, valuable products from recycled cellulose; such products include composite electrodes [4], bio-oil [5] and carbon dots [6].

A proportion of recycled fibres are used to produce new paper products in order to reduce environmental load and make better use of natural resources [7]. The collected paper materials are pulped, subjected to various chemical treatments and manufactured into boards or rolls, which are used to produce the products, primarily packaging materials. Paper is one of the most recyclable products and shows a global recycling rate of 72%.

The requirements of paper and paper-based products are changing [8]. The requirements for paper packaging have changed due to global marketing and improvements in graphic art and subsequent printing. Increased shipping distances require stronger and more durable packaging than was previously necessary. Additionally, economic considerations require pulp and paper makers to increase their levels of productivity, as with all other manufacturing industries [9]. This is achieved through the use of faster paper-making machines, which in turn require paper of increased strength and therefore an improvement in pulp-fibre properties.

The complexity of the problem means that careful study is required regarding the wetting characteristics [10] of the surface of corrugated paper that is selected for a specific use. To the best of our knowledge, no studies have addressed using an environmentally driven perspective to understand the surface characteristic of recycled cellulosic fibres

from arbitrary sources. All the experimental methods that were used in this study are adaptable to the study of wetting of the surfaces of lignocellulosic recycled fibres by different liquids. The corrugated medium, as the central layer of a board, is not subjected to surface modification or quality control and has not been studied from a surface energy point of view. Water and diiodomethane liquids were selected as the wetting agents in this analysis. The wetting mechanism varies according to the polarity of the agent or the presence of any surface-active solutes.

This paper describes the determination of the dispersive and polar components of the surface free energy (SFE) of recycled paper through the interaction of water and diiodomethane on industrially manufactured paper products.

## 2. Materials and Methods

Single wall, B-type (3 mm board thickness, 2.5 mm flute height, 6.1 mm wave length) commercial corrugated paperboard middle component paper layer (known as the fluting) was obtained in an undamaged and uncoated form from a Hungarian manufacturing company as a substrate material. The sheets were stated to be of grammage 80 $g/m^2$ and 100 $g/m^2$ and had been made during different manufacturing periods (124 samples were obtained for each type). They comprised 100% recycled fibres. The papers from which they were made had been collected within a six-month period. Corn-based starch adhesive was used in both types of sheets during manufacturing.

Two model probes were used: water that had been distilled twice in the laboratory and diiodomethane (Sigma-Aldrich). All products were used as received.

The ash content of the paper was 3.2%. The surface roughness for the 80 $g/m^2$ and 100 $g/m^2$ sheets was 5.24 and 4.83 μm, respectively. The surface roughness was evaluated using a Parker Print Surf instrument according to the T555 standard. A HITACHI S-3400N instrument was used for the scanning electron microscopy (SEM) imaging of the recycled paper cellulose sheet. The images were obtained at an operating voltage of 25 kV in backscattered electron mode. Digital Pocket optical microscopy was used for the optical imaging of the recycled paper surface.

### 2.1. Calculation of Contact Angle

Contact angles were measured through the use of a PGX goniometer (Sweden) according to the Tappi 458 standard. Calibrated liquid drops (5 μL as optimized value) were placed automatically on the sheet surface, while sessile and advancing drops, which were used for contact angle measurement, were illuminated by a diffuse light source. The contours of the drops were recorded through the use of a built-in camera. Then, special software was applied to calculate the advancing contact angle automatically in two modes: static (at specific time intervals) or dynamic (continuously in the given time frame). Five measurements were averaged for each grammage of paper.

### 2.2. Calculation of SFE

The SFE was calculated based on the sum of acid–base polar ($\gamma^p$) and Lifshitz–van der Waals non-polar ($\gamma^d$) interactions [11]. Deionised water acted as the polar and non-polar (dispersive) probe (since 63.7% of the total surface tension of water is polar), and the diiodomethane was considered as a non-polar only ($\gamma_l^p = 0$) dispersive liquid. Polar interactions can show an electron donor $\gamma_l^-$ and/or electron acceptor $\gamma_l^+$ character [12] due to hydrogen bonding. These interactions were calculated as a sum in the polar component. In diiodomethane, only dispersive, van der Waals interactions occur among its symmetrical molecules.

Deionised water was supplied from an inner container in the goniometer, while diiodomethane was supplied from a syringe that was separate from the machine. Diiodomethane liquid was injected from a 5 mL plastic syringe into a plastic tube, which was connected to the goniometer. A 5 μL drop of either agent could be placed on the surfaces of the papers.

To calculate the contribution of polar and dispersive components to the SFE of the recycled paper sheets, the following equation (Equation (1)) was used. This equation incorporates the Owens–Wendt approach, in which a term for hydrogen bonding is included [13]:

$$\gamma_l(1 + \cos\theta)/2 = \left(\gamma_s^d \gamma_l^d\right)^{1/2} + \left(\gamma_s^p \gamma_l^p\right)^{1/2} \tag{1}$$

In Equation (1), $\gamma_l^d$, $\gamma_s^d$ and $\gamma_l^p$, $\gamma_s^p$ are the dispersive and polar surface tension and SFE components of the liquids and solids, respectively. The surface tension of distilled water was estimated through the application of Equation (2):

$$\gamma_l = 235.8 \left(\frac{374 - T}{647.15}\right)^{1.256} \left[1 - 0.625\left(\frac{374 - T}{647.15}\right)\right] \tag{2}$$

In Equation (2), $T$ (°C) is the temperature of the distilled water.

The surface tension of diiodomethane was estimated through the application of the following equation:

$$\gamma_l = 53.48 - 0.14154\,T + 4.9567 \cdot 10^{-5} T^2 \tag{3}$$

in which $T$ (°C) is the temperature of the diiodomethane.

The test was performed in an air-conditioned room that was set at a temperature of $23 \pm 2$ °C and 50% relative humidity.

Based on the calculated and known SFE and surface tension of the solid and liquid, respectively, the immersion or interfacial free energy $\gamma_{sl}$ can be calculated as follows:

$$\gamma_{sl} = \gamma_l - 2\left[\left(\gamma_l^d \gamma_s^d\right)^{1/2} + \left(\gamma_l^p \gamma_s^p\right)^{1/2}\right] \tag{4}$$

If $\gamma_{sl} < 0$, it is thermodynamically a favourable system. The work of water that spreads on the recycled paper surface ($W_s$) was calculated through the use of the difference between the work of adhesion ($W_a$) and the work of cohesion ($W_c$) of water:

$$W_s = W_a - W_c = 2\left[\left(\gamma_s^d \gamma_l^d\right)^{1/2} + \left(\gamma_s^p \gamma_l^p\right)^{1/2}\right] - 2\gamma_w \tag{5}$$

The SFE parameters of the two probe liquids used in these experiments are listed in Table 1.

**Table 1.** Surface free energy components of the liquids used for the contact angle measurements at 23 °C.

| Liquid Probes | Surface Free Energy Parameters (mN/m) | | |
|---|---|---|---|
| | $\gamma_l^p = 2(\gamma_l^+ \gamma_l^-)^{1/2}$ | $\gamma_l^d$ | Total |
| water | 46.4 | 25.9 | 72.3 |
| diiodomethane | 0 | 50.2 | 50.2 |

## 3. Results

The contact angles of the test liquids on the corrugated paperboard sheets are given in Table 2.

**Table 2.** Contact angles of test liquids on the lignocellulose sheet surfaces.

| Corrugated Paperboards Middle Component Paper Samples | Contact Angle (°) | |
|---|---|---|
| | $\theta_{water}$ | $\theta_{diiodomethane}$ |
| 80 g/m² | 72.3 (17%) | 24.1 |
| 100 g/m² | 77.1 (17%) | 17.9 |

The standard deviation is reported in the brackets.

Slight variations were observed in the contact angles for the two different sheets. The higher grammage sheets showed higher angle values than the less dense sheets, as more fibres would be present in the same surface area, which would increase the surface polarity. No significant differences were observed in terms of the contact angle between the sheets that had been manufactured on different dates. The contact angles and standard deviations that are shown in Table 2 are average values for the 124 different production samples. In general, water comes into contact with paper sheets in several ways, and when it is absorbed, it disrupts the hydrogen bonding of chemically similar cellulosic fibre networks. Water can be stored in the external walls of the fibre structure and trapped in the pores among the fibres. Fibre lumen can also hold water, but in this material, the lumen is not considered to play a significant role because recycled fibres are flattened and the inner pores collapse due to the several recycling processes (press and drying) that are involved in the paper production. However, the external fibrillar structure absorbs water and becomes swollen, which loosens the network structure. Hence, the contact angle values are functions of the physical and chemical surface characteristics of the fibres.

From the contact angle values, the SFE of the samples and their dispersive and polar components were calculated according to the Owens–Wendt approach. The total SFE was approximately 49.5 mN/m and this energy was composed of high dispersive and low polar component values (Table 3). The total SFE for pure cellulose has been reported as 56.7 mN/m [14] and 48.11–57.11 mN/m [15]. These figures are much higher than those reported for recycled cellulose fibres. The SFE of laboratory-recycled cellulose fibre without any contamination has been reported to be 41.7 mN/m [16]. However, it is almost impossible to measure the SFE of intact recycled cellulose fibres as the product is always contaminated with chemicals or additives. The process by which corrugated board is manufactured does not involve cleaning of the fibres; moreover, additives, usually corn starch, are added to improve the mechanical properties of the products. The interfacial energy or SFE depends only on the individual properties of the test liquid and the quality of the recycled cellulose fibres.

**Table 3.** Surface free energies of the lignocellulose sheet surfaces.

| Corrugated Paperboards Middle Component Paper Samples | Surface Free Energy (mN/m) | | |
|---|---|---|---|
| | $\gamma_l^p$ | $\gamma_l^d$ | **Total** |
| 80 g/m$^2$ | 3.5 (23%) | 45.9 (4.2%) | 49.4 (4.8%) |
| 100 g/m$^2$ | 1.9 (21%) | 47.6 (2.5%) | 49.6 (2.5%) |

The standard deviation is reported in the brackets.

From consideration of the polar component values in Table 3, it can be concluded that the surface of recycled, contaminated cellulose fibre (depicted in Figure 1) does not interact greatly in a polar manner with water. The polar component, which arises due to hydrogen bonds, was found in this study to be 72.3% and 81.5% lower for the 80 g/m$^2$ and 100 g/m$^2$ samples, respectively, than was measured on a poly(tetrafluoroethylene) standardised test surface (46.4 mN/m). Because these polar surface tension and energy fractions are relatively far from each other, the cationic starch-treated recycled paper surface is less compatible with the water and the interfacial free energy is also not minimized. To produce a paper surface that can be glued well, the paper and adhesive should show similar cohesive interaction, so an adhesive with a low polar surface tension component should be selected or the paper surface would show increased polarity (hydrophilicity).

Diiodomethane interacts with the recycled paper surface only through dispersive forces (non-site-specific van der Waals forces) as it has molecular symmetry and does not contain heteroatoms. Compared with the polar surface component, the dispersive loss is around 50% and this reduced loss results in the production of a stable sessile diiodomethane drop on the paper surface. Similar data was reported by Kondor et al. in 2021 [17]. Diiodomethane has a higher $\gamma_l^d$ value than water, so its interaction intensity with the recycled fibre network is expected to be higher than that with pure water.

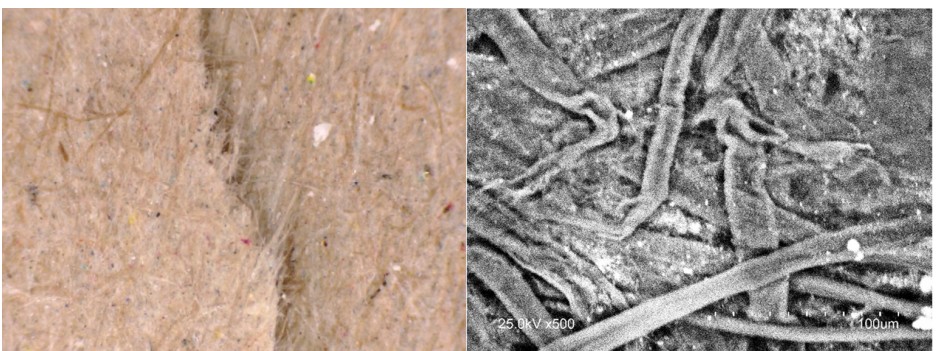

**Figure 1.** Optical (**left**) and SE (**right**) microscopic images of the corrugated paperboard middle component paper layer.

The dispersive component of the SFE is caused by an instantaneous dipole moment, which is similar to the Coulomb interaction between an electron and the nuclei in two molecules [18]. In this study, the total decrease in SFE that was found with the higher grammage paper implies a higher contact angle value for the water compared with that observed for the lower grammage paper. A higher contact angle indicates weaker hydrogen bonding, which results in a lower SFE. The SFE is a direct consequence of asymmetrical force distribution between atoms or molecules at the interface and in the bulk material. The above results indicate a good correlation between the SFE and the surface composition of the recycled fibres.

A drastic decrease was observed in the contact angle values within three seconds of the fall of a drop. After this, a gradual decrease was observed during the rest of the 10 s observation period. The water contact angles were much lower than those of diiodomethane. This behaviour can be interpreted as being due to the saturation of the surface of the hydrophilic lignocellulose fibre.

Diiodomethane molecules are three times larger than those of water [19]. Therefore, water molecules can disperse easily on the fibre surface and penetrate the recycled fibre network due to the presence of polar components on the fibre surfaces. Moreover, the contact angle of diiodomethane changed less than that of water.

The entropy (wetting energy) of adsorption was found to be lower than zero ($-16.2$ mJ/m$^2$ and $-21.8$ mJ/m$^2$ for the 80 g/m$^2$ and 100 g/m$^2$ sheets, respectively) for the investigated liquid systems and it decreased with the strength of the interaction forces between the surface of the recycled fibres and the water.

## 4. Conclusions

To summarise, the differences in the contact angles that were measured within the samples indicate that these values are very sensitive to the quality of the raw material and the ordering and packing of charged molecules or impurities on the fibre surfaces. Recycling does not change the overall total SFE but it can increase both the dispersive and the polar components.

**Author Contributions:** Conceptualization, L.C. and J.K.; methodology, L.C.; investigation, J.K.; resources, J.K.; data curation, J.K.; writing—original draft preparation, L.C.; writing—review and editing, L.C. All authors have read and agreed to the published version of the manuscript.

**Funding:** This research received no external funding.

**Institutional Review Board Statement:** Not applicable.

**Informed Consent Statement:** Not applicable.

**Data Availability Statement:** Not applicable.

**Conflicts of Interest:** The authors declare no conflict of interest.

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
