# Peer review of "Characterisation of the Surface Free Energy of the Recycled Cellulose Layer that Comprises the Middle Component of Corrugated Paperboards"

_coatings, doi:10.3390/coatings13020259_

Round 1
Author Response
The manuscript was revised according to this Reviewer’s suggestions. In the text of the revised manuscript the changes made are marked in red.
We thank the Reviewer for the encouraging comments on the manuscript. The manuscript has been modified and corrected according to this Reviewer’s suggestion.
Comment: In page 2, paragraph 2, line 54, what type of corrugated board do you mean? do you mean single wall board? you need to add it precisely to the text.
Answer: The final product was a single-wall, B-type corrugated paperboard. The information has been added to Line 59.
Comment: In page 2, paragraph 2, line 59, what was the amount of applied Corn based starch?
Answer: The industrial partner did not give us information about the details of the adhesive, as this is confidential information.
Comment: I think you must have a separate material data table (physical properties) for fluting of the corrugated board, including direction (MD or CD), thickness, flute height.
Answer: The information has been added to Line 59.
Comment: In this paper there is not any morphology results related to the flute paper or maybe corrugated board, I think it is always so important to show them, please insert SEM images or optical microscopy images.
Answer: Optical and SE microscopy images have been added to the manuscript to depict the contaminated corrugated paperboard flute surface.
Comment: One of the important factors that can affect to the contact angle and surface tension results is surface roughness, please provide the data.
Answer: Parker Print Surf surface roughness of the recycled papers has been evaluated. The roughness value for the 80 and 100 g/m^2 sheets are 5.24 and 4.83 μm respectively. The information has been added to the manuscript.
Reviewer 2 Report
Dear Authors,
Please follow the comments.
Good luck
Line 13 and 17, please add (s) to the words of fiber and product.
Line 34, please add (s) to the word of requirement.
Line 39, please change the word of increase to this one (increasing).
In the last paragraph of introduction, you have to express the research gap and more details that why this research is important. Thus, please add at least two new paragraphs. One paragraph for comparing the previous articles, and the other for your work and the importance of your work.
Line 59 and 60, please add more details of your adhesive which was used (such as density of the adhesive, solid content, and viscosity).
Line 81, why did you use the amount of the liquid (5)? Did you use any standards in terms of the number?
After table 1, where is section of data analysis of your work? You did not use any software in order to check?
Line 112, the dictation of the word (distillated) is wrong. Please correct it.
In table 2 and 3, please remove (%) in the brackets.
Line 121, please remove the (s) from shows.
Line 122, please add (s) to the increase.
Please revise all the English manuscript of your paper after line 122. I will not correct any grammatical of dictation of the manuscript.
On page five, please specify the section of conclusion and expanded more.
Author Response
The manuscript was revised according to this Reviewer’s suggestions. In the text of the revised manuscript the changes made are marked in red.
We thank the Reviewer for the encouraging comments on the manuscript. The manuscript has been modified and corrected according to this Reviewer’s suggestion.
Comment: Line 13 and 17, please add (s) to the words of fiber and product.
Answer: The final product was a single-wall, B-type corrugated paperboard. The recycled fibres originated from a different source but did not receive more information on them. It has been added to Line 59.
Comment: Line 34, please add (s) to the word of requirement.
Answer: The whole manuscript underwent proofreading by a paid service and was completely rewritten.
Comment: Line 39, please change the word of increase to this one (increasing).
Answer: The whole manuscript underwent proofreading by a paid service and was completely rewritten.
Comment: In the last paragraph of introduction, you have to express the research gap and more details that why this research is important. Thus, please add at least two new paragraphs. One paragraph for comparing the previous articles, and the other for your work and the importance of your work.
Answer: The necessary changes have been made in the manuscript.
Comment: Line 59 and 60, please add more details of your adhesive which was used (such as density of the adhesive, solid content, and viscosity).
Answer: The industrial partner did not give us information about the details of the adhesive, as this is confidential information.
Comment: Line 81, why did you use the amount of the liquid (5)? Did you use any standards in terms of the number?
Answer: Tappi 458 standard has been used. The 5 μl drop size was selected after optimization. The necessary changes have been added to the text.
Comment: After table 1, where is section of data analysis of your work? You did not use any software in order to check?
Answer: The values have been calculated manually according to the given equations, and no software has been used.
Comment: Line 112, the dictation of the word (distillated) is wrong. Please correct it.
Answer: Thank you, it has been rephrased.
Comment: In table 2 and 3, please remove (%) in the brackets.
Answer: We think that (%) is important there.
Comment: Line 121, please remove the (s) from shows.
Answer: The whole manuscript underwent proofreading by a paid service and was completely rewritten.
Comment: Line 122, please add (s) to the increase.
Answer: The whole manuscript underwent proofreading by a paid service and was completely rewritten.
Comment: Please revise all the English manuscript of your paper after line 122. I will not correct any grammatical of dictation of the manuscript.
Answer: The whole manuscript underwent proofreading by a paid service and was completely rewritten.
Comment: On page five, please specify the section of conclusion and expanded more.
Answer: The conclusion section has been separated.
Round 2
Reviewer 2 Report
Dear Author,
The manuscript has now been better than former version.
Good Luck